# Colloid Carcinoma of the Pancreas with a Series of Radiological and Pathological Studies for Diagnosis: A Case Report

**DOI:** 10.3390/diagnostics12020282

**Published:** 2022-01-22

**Authors:** Chuan-Han Chen, Hong-Zen Yeh, Hsin-Ni Li

**Affiliations:** 1Department of Radiology, Taichung Veterans General Hospital, Taichung 40705, Taiwan; bonanza0622@gmail.com; 2Division of Gastroenteroenterology and Hepatology, Department of Internal Medicine, Taichung Veterans General Hospital, Taichung 40705, Taiwan; hzen.yeh@gmail.com; 3College of Medicine, National Yang Ming Chiao Tung University, Taipei 11221, Taiwan; 4Division of Gastroenterology and Hepatology, Tungs’ Taichung Metroharbor Hospital, Taichung 435403, Taiwan; 5Department of Pathology and Laboratory Medicine, Taichung Veterans General Hospital, Taichung 40705, Taiwan; 6Department of Nursing, National Taichung University of Science and Technology, Taichung 40343, Taiwan

**Keywords:** colloid carcinoma (CC) of the pancreas, mucinous cystic neoplasm (MCN), intraductal papillary mucinous neoplasm (IPMN)

## Abstract

Pancreatic colloid carcinoma is an uncommon and unique malignancy possessing a significantly more favorable prognosis than that of ordinary pancreatic ductal adenocarcinoma. Accurate diagnosis of this rare entity is thus important for leading the ensuing optimal treatment. Herein we report a case of colloid carcinoma of the pancreas with a series of imaging findings and pathologic assessments. Being familiar with these radio-pathological features makes early diagnosis possible prior to operation.

## 1. Introduction

Colloid carcinoma (CC) of the pancreas, also known as mucinous non-cystic carcinoma, is a rare subtype of pancreatic cancer, accounting for only 1–3% of the malignant neoplasms of the exocrine pancreas [1,2]. It is evident that pancreatic CC may arise in association with an intestinal-type intraductal papillary mucinous neoplasm (IPMN) or mucinous cystic neoplasm (MCN) [1,3,4]. During imaging, a lobulated mass with indiscrete margins may be seen. The mass presents a cystic appearance due to mucin production and heterogeneous enhancement based upon its histopathologic components [5]. Histomorphologically, the tumor is characterized by well-delineated extracellular mucin pools, which contain detached floating clusters of neoplastic cells [1,2,4]. Identifying this unique variant of pancreatic cancer is important as it has a significantly more favorable prognosis than that of conventional ductal adenocarcinoma of the pancreas [1]. In addition, treatment modalities are different as surgical intervention alone may be sufficient for the management of node-negative CC, while in ordinary pancreatic ductal adenocarcinoma, administrating systemic chemotherapy is usually recommended [6]. However, it is difficult to make a confident diagnosis of CC preoperatively because of the rarity of this entity and the dearth of experience, particularly in cases where the imaging findings are not compatible with the most reported imaging features and the limited amount of biopsied tissue. Herein we report a case of CC with a series of imaging studies and pathological assessments. Through the use of radiological-pathological correlation, the correct diagnosis may therefore be developed preoperatively.

## 2. Case Presentation

A 46-year-old female with a medical history of type II diabetes mellitus and idiopathic thrombocytopenia had regular follow-ups at the Department of Hemato-oncology at Taichung Veterans General Hospital, Taiwan. Due to her persistent thrombocytopenia in spite of therapy, imaging studies for an abdominal survey were arranged (Figure 1A–D). On computed tomography (CT) of the abdomen, a well-defined lobulated cystic mass was noted in the pancreatic head, where there was the presence of peripheral and internal calcifications and faint enhancement. Some smaller lesions having similar patterns were found around the mass abutting the pancreatic head. These imaging findings were suggestive of a cystic neoplasm of the pancreas, presumably malignant in nature with metastases involving lymph nodes. After that, the patient underwent magnetic resonance cholangiopancreatography (MRCP) for further evaluation. The cystic mass showed generally high signal intensity with some low signal foci and internal septa on T2-weighted images (T2WI). On dynamic imaging, the mass showed gradual enhancement at the periphery and internal septa. Some foci of faint enhancement were found within the cystic parts of the mass. In addition, possible metastases with enlarged lymph nodes were also found around the pancreatic head. Neither dilatation of the pancreatic duct nor communication between the mass and the pancreatic duct was found. A malignant mucinous cystic neoplasm of the pancreas with metastases involving lymph nodes was suspected, while IPMN was less likely radiologically.

Endoscopic ultrasound (EUS) revealed a heterogeneous hypoechoic lesion with a cystic component around the pancreatic head, measuring 50.1 mm× 40.4 mm in size, in the presence of calcifications (Figure 2A,B). Duodenal wall thickening with ulceration was also found.

EUS-guided aspiration of the pancreatic lesion and a biopsy of the duodenal ulceration reported adenocarcinoma in a background of copious extracellular mucin (Figure 3A–D). Prior to surgery, the patient received neoadjuvant chemotherapy using FOLFIRINOX (folinic acid, fluorouracil, irinotecan and oxaliplatin) because conventional pancreatic adenocarcinoma arising from a mucin-rich neoplasm was impressed. The follow-up imaging study 10 months later revealed a slight increase in size and more calcifications in the mass and the metastases with enlarged lymph nodes. Subsequently, the patient underwent a pylorus-preserving pancreaticoduodenectomy and regional lymph node dissection. During surgery, a 6 cm × 5 cm multilocular cystic tumor was observed in the pancreatic head, with invasion to the duodenum. Lymphadenopathies were found behind the pancreatic head (group 13) and at the superior mesenteric artery (group 14).

Histopathological analysis of the resected specimen revealed a 60 mm× 50 mm multilocular cystic lesion at the head of the pancreas, with involvement of the duodenum. Microscopically, the tumor displayed the presence of abundant extracellular mucin containing scanty neoplastic epithelial cells. The malignant epithelium floating within the mucin was arranged in strips, clusters, glands and as individual cells. Some floating cells showed signet-ring type features (Figure 3E–G). There was a strong expression of CDX2 (Figure 3H) and increased expression of p53 on these neoplastic cells (Figure 3I). Prominent calcification was observed in the mucin pools. All the resection margins were negative for neoplastic cells. In terms of metastasis in the resected lymph nodes, lymph node involvement was observed to have an identical morphology as the pancreatic tumor. According to the *American Joint Committee on Cancer* tumor-node-metastasis staging of pancreatic cancer (8th edition, 2017), the tumor was assigned as pathological stage T3N1 (stage IIB).

During the postoperative period, the patient was treated with adjuvant chemotherapy (Gemcitabine). The patient exhibited neither evidence of recurrence nor newly-developed metastases after 30 months of follow-up.

## 3. Discussion

CC of the pancreas attains a relatively protracted clinical course compared with that of conventional ductal adenocarcinoma of the organ. The 5-year survival rate of pancreatic CC was found to be 57%, in contrast with the 12% survival rate in resectable cases of conventional adenocarcinoma. Lymph node metastases occur in 30–40% of cases. Even in the presence of lymph node metastases, patients with pancreatic CC still have a favorable prognosis [1,2]. Due to the dissimilar, biological and clinical behaviors from the fatal ordinary pancreatic cancer, an early and accurate diagnosis is pivotal to provide appropriate management for the patients.

Only a few reports focusing on imaging findings of the CC of the pancreas have been documented. Yoon et al. described the condition as a mass with a lobulated contour and indiscrete margins. During T2WI of magnetic resonance imaging (MRI), it may show a salt-and-pepper appearance, reflecting the histopathologic composition of mucin pools, intervening fine stroma, and the neoplastic cells. During dynamic imaging, progressive delayed peripheral and internal sponge-like or mesh-like enhancement of the intervening stroma may be seen [5]. Jiang et al. observed that calcifications were found in certain cases (3 out of 5) with different patterns, while only one case had calcifications within the tumor [7]. The imaging findings in our case were mostly compatible with the aforementioned descriptions of CC except for a relatively circumscribed margin, as well as internal and peripheral calcifications of the lesion. Differential diagnoses included both IPMN and MCN when encountering a cystic neoplasm of the pancreas, as was seen in this case. Typical imaging characteristics of IPMN, such as communication between the mass and pancreatic duct, downstream dilatation of the pancreatic duct or intraductal papillary components [8], were not seen in this case. In terms of MCN, this neoplasm usually presents itself as a single spherical mass with thick septae and peripheral calcifications, typically located at the body and the tail of the pancreas [9]. This is different from our case, which possessed a lobulated contour at the pancreatic head in the presence of internal and peripheral calcifications. In addition, Yoon and Jiang also mentioned lymph node involvement in certain cases within their case series but did not show imaging appearances in their papers. Enlarged lymph nodes were noticed in our case, with identical imaging patterns as the main tumor, indicating the possibility of lymph node involvement. Taken together, these imaging findings may hint that the case presented was an unusual malignant cystic neoplasm of the pancreas with lymph node metastases.

Pathologically, distinguishing CC from other pancreatic tumors with intraluminal mucin deposition, such as IPMN and MCN is also difficult, particularly in aspiration or small biopsied specimens. The fact that CC often coexists with these two mucin-rich neoplasms further complicates the distinction. Thus, it is important to be familiar with their morphologic pictures. In contrast with the stromal mucin, which firmly adheres to the tissue of colloid carcinoma, luminal mucin of IPMNs and MCNs is easily washed away during processing of the specimens, leaving scanty mucin for microscopic evaluation. In addition, the observation that the epithelial cells of colloid carcinoma are loosely attached to the stroma and mostly float within the mucin while the epithelial cells continuously line the involved cysts or ducts of MCNs and IPMNs, make the differentiation possible [1,2,4]. Moreover, the recent advancement of tools, such as EUS-guided through-the-needle micro-forceps for use in pancreatic cystic wall biopsies, may improve diagnostic accuracy in biopsied specimens by obtaining characteristic stroma of cystic lesions of the pancreas [10].

In our case, we noticed that scattered neoplastic cell clusters and glands were suspended in a background of large mucin pools in the absence of contiguous epithelial lining, most suggestive of the CC of the pancreas. The initial pathology report described the lesion as an adenocarcinoma in a background of abundant extracellular mucin, which did not inform the clinical doctors of the possibility of pancreatic CC. Neoadjuvant chemotherapy was thus administered as conventional pancreatic carcinoma arising from mucin-producing neoplasm had been impressed. The combination of radiologic and pathologic features (Table 1) makes the diagnosis of CC more certain at the onset and can help guide the medical team to follow the appropriate treatment.

A national database-based study suggested that surgery alone may be sufficient for the management of node-negative (I/IIA) CC, in contrast to conventional pancreatic ductal adenocarcinoma, whereas CC patients with stage IIB disease had a survival benefit from perioperative chemoradiation [6]. However, the role that neoadjuvant chemotherapy plays is unknown in this unique variant of pancreatic neoplasm. Neoadjuvant chemotherapy is often used in the management of borderline resectable pancreatic cancer to both reduce tumor size and increase the resectability and margin-negative resection rate [15]. To the best of our knowledge, no literature exists regarding the use of neoadjuvant chemotherapy for this type of tumor. In our case, we noted that there was a persistent mass with low-cellular mucin pools and prominent calcification after neoadjuvant chemotherapy, which may be compatible with the findings of invasive mucinous carcinoma of the breast, a morphological similarity seen in other organs treated with neoadjuvant chemotherapy [16]. Discordant clinical, imaging and pathological findings in terms of disease progression or regression may occur in the setting of post-neoadjuvant chemotherapy in this case.

## 4. Conclusions

We have reported a case of a rare subtype of pancreatic cancer, colloid carcinoma. The combination of both radiologic and histopathologic findings can help direct medical personnel towards an early and accurate diagnosis, as well as help guide the subsequent appropriate management.

## Figures and Tables

**Figure 1 diagnostics-12-00282-f001:**
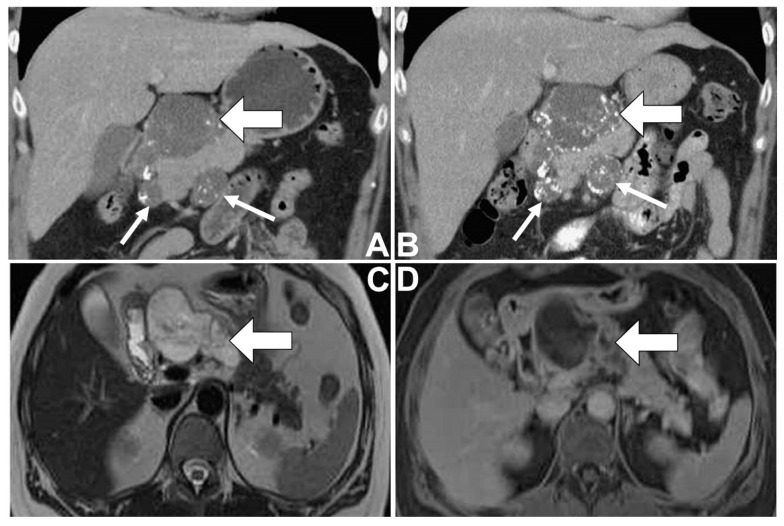
Colloid carcinoma of the pancreas in a 46-year-old woman. The coronal contrast-enhanced CT images at the time of initial presentation (**A**) and 10 months later (**B**) show a large well-defined lobulated cystic-appearing mass in the pancreatic head (thick arrow). Scattered peripheral and internal calcifications of the mass are noted. Lymphadenopathies (thin arrows) with similar imaging patterns of the mass are found around the pancreatic head. The follow-up image shows a slight increase in lesion size and calcifications. Marginal and septal enhancement and some intracystic enhancing foci are also found. The MRI at the time of initial presentation reveals a lobulated circumscribed hyperintense mass with some hypointense septa and foci on axial T2WI (**C**), as well as compatible CT enhancing patterns on axial T1WI gadolinium-enhanced delayed phase of MRI (**D**).

**Figure 2 diagnostics-12-00282-f002:**
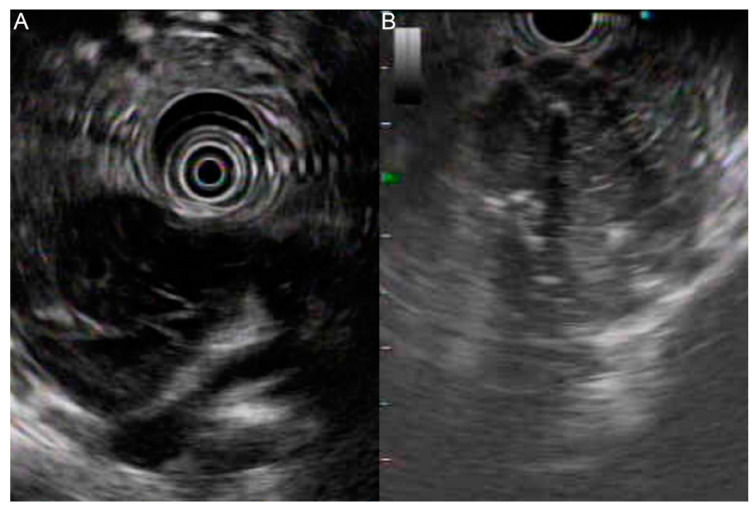
EUS demonstrating a heterogeneous hypoechoic lesion with a cystic component at the pancreatic head (**A**) in the presence of calcifications (**B**).

**Figure 3 diagnostics-12-00282-f003:**
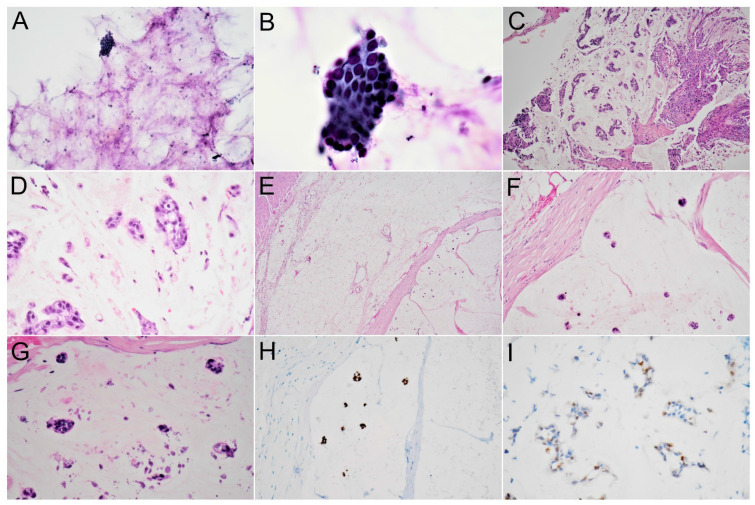
Cytology and histology of colloid carcinoma of the pancreas. (**A**) Needle aspiration cytology of the pancreatic tumor shows scanty floating glands in the background of extracellular mucin pools (×200). (**B**) High magnification reveals atypical glands in a mucinous background (oil immersion, ×1000). (**C**) Duodenal biopsy reveals scattered glands or strips of epithelial cells in a background of copious extracellular mucin with surface ulceration (H&E stain, ×100). (**D**) Cellular clusters and single-cell with signet-ring features (H&E stain, ×200). (**E**) The tumor invades the periduodenal soft tissue in the presence of a thick fibrous wall (H&E stain, ×40). (**F**,**G**) High magnification reveals scanty glands and prominent calcification in the extracellular mucin pools (H&E stain, ×100 and ×400). (**H**) Immunohistochemistry of CDX2 highlights neoplastic cells (×200). (**I**) Increased expression of p53 immunohistochemistry stain is seen in neoplastic cells (×200).

**Table 1 diagnostics-12-00282-t001:** Distinguishing clinical, radiological and pathological features of mucin-rich neoplasms of the pancreas [2,9,11,12,13,14].

Features		IPMN	MCN	CC/Our Case
Clinical	Age	Elderly	Middle	Middle to elderly/middle
	Gender	Male (70%)	Female (95%)	No gender predominance/female
Radiological	Location	Head (70%)	Body and tail (95%)	Head/head
	Shape	Ovoid	Spheroid	Lobulated/lobulated
	Duct communication	Common	No	No/No
	Calcification	Some (20%)	Some (30%) (peripheral)	Some/Yes (internal and peripheral)
	MRI T2 high signal intensity	Yes	Yes	Yes/Yes (salt-and-pepper appearance)
Pathological	Stromal mucin in biopsied specimens	Scanty	Scanty	Abundant/mucin pools
	Epithelial lining	Continuous columnar or papillary	Continuous columnar	Cell clusters floating within the mucin/Cell clusters or strips in mucin pools
	Stroma	Fibrotic	Ovarian	Fibrotic/Fibrotic

Abbreviation: CC, colloid carcinoma; IPMN, intraductal papillary mucinous neoplasm; MCN, mucinous cystic neoplasm.

## Data Availability

Not applicable.

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
