# Peer review of "Colloid Carcinoma of the Pancreas with a Series of Radiological and Pathological Studies for Diagnosis: A Case Report"

_diagnostics, 2022, doi:10.3390/diagnostics12020282_

Round 1
Reviewer 1 Report
A very interesting and well-presented case of a rare cystic invasive tumor of the pancreas.
To further improve this paper, I suggest:
- include some EUS images if available
- Discuss the role of new tools for cystic wall biopsy (microforceps) with specific comment on the possibility to obtain histological specimens including stroma (i.e., ovarian-like stroma). Doing so cite PMID: 29899433
Author Response
Response to Reviewer 1 Comments:
A very interesting and well-presented case of a rare cystic invasive tumor of the pancreas.
Response: Many thanks for the compliments.
To further improve this paper, I suggest:
- include some EUS images if available
Response 1: Many thanks for the useful suggestions. We have added the EUS image which is Fig. 2. EUS findings have been also briefly described.
- Discuss the role of new tools for cystic wall biopsy (microforceps) with specific comment on the possibility to obtain histological specimens including stroma (i.e., ovarian-like stroma). Doing so cite PMID: 29899433
Response 2: Many thanks for the useful suggestions. Advanced tools for the biopsies of pancreatic cystic lesions such as EUS-guided through-the-needle micro-forceps have been mentioned in the Discussion Section (Line 176-178) and the article you suggested is cited accordingly.
Reviewer 2 Report
Line 41: change woman to female
Line 43: please clarify: “ thrombocytopenia even under medication” do you in spite of therapy?
Line 55: “metastatic lymphadenopathies”- reword with ‘possible metastases with enlarged lymph nodes’.
Line 57: “metastatic lymphadenopathies” This term is incorrect- reword as above
Line 62: “metastatic lymphadenopathies” adenopathy
Line 63: Remove word “simultaneously”
Figure 1 is missing a description for image D. If it is the same as C the change to (C,D)
Line 68: “Endoscopic ultrasound-guided aspiration and biopsy of duodenum.” Please clarify, was the lesion biopsied through a duodenal approach or was the duodenum itself biopsied?
Line 87: “staging of pancreatic cancer (year)” What is the (year) or why is it included?
Line 90 “histomorphology” change to “histology”
Figure 1, 2- use the same formatting to describe the figures either A,B, or (A) (B).
Line 95: F and G please comply to formatting of all Figure legends (F) (G)
Line 98: “At the post-operative follow-up, the patient was followed with adjuvant chemother..” Change to During the post-operative period the patient was treated with adjuvant chemother
Line 99: use only generic names (Gemzar) change to gemcitabine
Line 108: “fatal ordinary pancreatic cancer” change to adenocarcinoma
Line 110: “Only few reports have been published about imaging findings of CC of, describing it as a mass
Line 120: “it usually presented”…change to “this neoplasm usually presents”
Line 147: “there is no literature about neoadjuvant chemotherapy application on” change to ‘there is no literature regarding the use of neoadjuvant chemotherapy for”
The paper would be improved by reviewing other cases in the literature (there are several) and comparing their findings to others.
Author Response
Response to Reviewer 2 Comments:
Response: Many thanks for all the suggestions for the formatting and language revisions. The terms, phrases and formatting have been changed as recommended.
Line 41: change woman to female
Response 1: The term has been changed accordingly.
Line 43: please clarify: “ thrombocytopenia even under medication” do you in spite of therapy?
Response 2: The term has been changed to “in spite of therapy”.
Line 55: “metastatic lymphadenopathies”- reword with ‘possible metastases with enlarged lymph nodes’.
Response 3: The term has been changed accordingly.
Line 57: “metastatic lymphadenopathies” This term is incorrect- reword as above
Response 4: The term has been changed accordingly.
Line 62: “metastatic lymphadenopathies” adenopathy
Response 5: The term has been changed accordingly.
Line 63: Remove word “simultaneously”
Response 6: The term has been removed accordingly.
Figure 1 is missing a description for image D. If it is the same as C the change to (C,D)
Response 7: Description D has been added.
Line 68: “Endoscopic ultrasound-guided aspiration and biopsy of duodenum.” Please clarify, was the lesion biopsied through a duodenal approach or was the duodenum itself biopsied?
Response 8: EUS-guided aspiration of pancreatic lesion and biopsy of duodenal wall were done and the specimens obtained were clarified in the revised manuscript.
Line 87: “staging of pancreatic cancer (year)” What is the (year) or why is it included?
Response 9: We are sorry for the missing typo. The edition and the year of AJCC published were added in the revised manuscript.
Line 90 “histomorphology” change to “histology”
Response 10: The term has been changed accordingly.
Figure 1, 2- use the same formatting to describe the figures either A,B, or (A) (B).
Response 11: The formatting has been changed to make it consistent through the figures.
Line 95: F and G please comply to formatting of all Figure legends (F) (G)
Response 12: The formatting has been changed to make it consistent through the figures.
Line 98: “At the post-operative follow-up, the patient was followed with adjuvant chemother..” Change to During the post-operative period the patient was treated with adjuvant chemother
Response 13: The phrase has been changed accordingly.
Line 99: use only generic names (Gemzar) change to gemcitabine
Response 14: The term has been changed accordingly.
Line 108: “fatal ordinary pancreatic cancer” change to adenocarcinoma
Response 15: The term has been changed accordingly.
Line 110: “Only few reports have been published about imaging findings of CC of, describing it as a mass
Response 16: The sentence has been rephrased.
Line 120: “it usually presented”…change to “this neoplasm usually presents”
Response 17: The phrase has been changed accordingly.
Line 147: “there is no literature about neoadjuvant chemotherapy application on” change to ‘there is no literature regarding the use of neoadjuvant chemotherapy for”
Response 18: The phrase has been changed accordingly.
The paper would be improved by reviewing other cases in the literature (there are several) and comparing their findings to others.
Response 19: Many thanks for the useful suggestions. Literature reviews to compare their findings with ours have been included in the second and fourth paragraphs of Discussion Section and a comparison table has been made.
Reviewer 3 Report
Chen and Li describe the case of a patient with colloid carcinoma oft he pancreas, a rare entity which is worthwhile reporting.
The title suggests ‚radiological and pathological studies for diagnosis‘, which not truly the case. In fact, the authors did apparently not recognize this entity before treatment, as suggested by the start treatment with systemic neoadjuvant chemotherapy.
The manuscript would profit from tables which compare a) the imaging and b) the pathologic features of colloid carcinoma in general with those of the colloid carcinoma described in this case as well as of its major differential diagnoses, namely IPMN and MCN. In fact, the diagnostic features which enable the diagnosis of CC in ths case are not described.
It remains unclear, why neoadjuvant treatment was started.
There are some typos:
- Page 2/70 FOLFIRINOX, not FORFIRINOX
- Page 3/87 ‚(year)‘
- Page 4/99 ‚Gemzar‘
Author Response
Response to Reviewer 3 Comments:
Chen and Li describe the case of a patient with colloid carcinoma oft he pancreas, a rare entity which is worthwhile reporting.
The title suggests ‚radiological and pathological studies for diagnosis‘, which not truly the case. In fact, the authors did apparently not recognize this entity before treatment, as suggested by the start treatment with systemic neoadjuvant chemotherapy.
Response 1: Many thanks for your suggestions. We have to apologize that we did not make it clear in our previous manuscript that the case shall be correctly diagnosed initially based on the characteristic radiological and pathological features. As the pathology report described the lesion as an adenocarcinoma in a background of extracellular mucin, which was misinterpreted by the clinicians that it was a conventional pancreatic adenocarcinoma with abundant mucin production. Thus systemic neoadjuvant was administered. The patient would not receive the controversial neoadjuvant chemotherapy if we had been familiar with this entity and made the diagnosis of colloid carcinoma more confidently prior to treatment. The reason why the patient received preoperative neoadjuvant chemotherapy has been added in the fourth paragraph of the Discussion Section in our revised manuscript.
The manuscript would profit from tables which compare a) the imaging and b) the pathologic features of colloid carcinoma in general with those of the colloid carcinoma described in this case as well as of its major differential diagnoses, namely IPMN and MCN. In fact, the diagnostic features which enable the diagnosis of CC in the case are not described.
Response 2: Many thanks for your suggestions. A table comparing IPMN, MCN and CC in regard to their clinical, radiological, and pathological features has been added in the revised manuscript.
It remains unclear, why neoadjuvant treatment was started.
Response 3: As I mentioned above, the initial pathology report did not inform the possibility of CC thus the patient was treated as the case of conventional pancreatic adenocarcinoma.
There are some typos:
Response 4: Many thanks for your kind remind. The typos have been changed accordingly.
Page 2/70 FOLFIRINOX, not FORFIRINOX
Page 3/87 ‚(year)‘
Page 4/99 ‚Gemzar‘